# E-cigarette adverts and children's perceptions of tobacco smoking harms: an experimental study and meta-analysis

Milica Vasiljevic,[1] Amelia St John Wallis,[1] Saphsa Codling,[1] Dominique-Laurent Couturier,[1] Stephen Sutton,[1,2] Theresa M Marteau[1]

[1]Behaviour and Health Research Unit, Institute of Public Health, University of Cambridge, Cambridge, UK
[2]Behavioural Science Group, Institute of Public Health, University of Cambridge, Cambridge, UK

**Correspondence to**
Dr Milica Vasiljevic;
milica.vasiljevic@medschl.cam.ac.uk

## ABSTRACT

**Objectives** Children exposed to electronic cigarette (e-cigarette) adverts may perceive occasional tobacco smoking as less harmful than children not exposed to e-cigarette adverts. Given the potential cross-cueing effects of e-cigarette adverts on tobacco smoking, there is an urgent need to establish whether the effect found in prior research is robust and replicable using a larger sample and a stronger control condition.

**Design** A between-subjects experiment with one independent factor of two levels corresponding to the advertisements to which participants were exposed: glamorous adverts for e-cigarettes, or adverts for objects unrelated to smoking or vaping.

**Participants** English school children aged 11–16 (n=1449).

**Outcomes** Perceived harm of occasional smoking of one or two tobacco cigarettes was the primary outcome. Secondary outcomes included: perceived harm of regular tobacco smoking, susceptibility to tobacco smoking and perceived prevalence of tobacco smoking in young people. Perceptions of using e-cigarettes were gauged by adapting all the outcome measures used to assess perceptions of tobacco smoking.

**Results** Tobacco smokers and e-cigarette users were excluded from analyses (final sample n=1057). Children exposed to glamorous e-cigarette adverts perceived the harms of occasional smoking of one or two tobacco cigarettes to be lower than those in the control group (Z=−2.13, p=0.033). An updated meta-analysis comprising three studies with 1935 children confirmed that exposure to different types of e-cigarette adverts (glamorous, healthful, flavoured, non-flavoured) lowers the perceived harm of occasional smoking of one or two tobacco cigarettes (Z=3.21, p=0.001).

**Conclusions** This study adds to existing evidence that exposure to e-cigarette adverts reduces children's perceptions of the harm of occasional tobacco smoking.

## INTRODUCTION

Fewer children are smoking tobacco cigarettes today than several decades ago. However, the advent of electronic cigarettes (e-cigarettes) could disrupt this trend. The availability and

### Strengths and limitations of this study

► We replicate and extend prior findings regarding the perceived harm of occasional tobacco smoking using a larger sample and a stronger control condition.
► Meta-analysis of three studies confirms that exposing children to different e-cigarette adverts (glamorous, healthful, flavoured, or non-flavoured) lowers their perceived harm of occasional tobacco smoking.
► The present study was limited in several respects: the primary outcome measured perceived risk of smoking, not behaviour; and the design used only momentary exposure to e-cigarette adverts.

use of e-cigarettes has risen rapidly in the last six years with an estimated 12%–24% of children aged 11–18 experimenting at least once with e-cigarettes in Great Britain in 2015/2016,[1] and 13.5% of middle schoolers and 37.7% of high schoolers in the USA in 2016.[2 3]

E-cigarettes have the potential for benefit and harm, the nature and scale of each being uncertain in the absence of much evidence. One potential benefit comes from providing a safe delivery mechanism for nicotine and an effective cessation aid. Evidence is accumulating to suggest that e-cigarettes can successfully be used as cessation aids by smokers.[4 5] Of concern, however, is their potential to make attitudes towards tobacco smoking more positive (ie, to renormalise it) through, for example, marketing of objects that appear very similar to tobacco cigarettes that appeal to both adult and children who are non-smokers. Any such impact on children is of particular concern given the potential for any changes in attitudes to tobacco smoking to increase the chances of tobacco smoking in this group in particular.[3 6 7]

Several prospective studies in the USA and UK have found that among children,

e-cigarette use predicts tobacco smoking one year later.[8–12] By contrast, population-level data show that the rising use and experimentation of e-cigarettes among children is accompanied by a continued decline in regular tobacco smoking in that group, from 15.8% to 8% among US high schoolers and from 4.3% to 2.2% among US middle schoolers in the period from 2011 to 2016,[2] and from 5% in 2011 to 3% in 2016 among 11–15 year olds in England.[13] Similar declines in rates of ever smoking tobacco (25% to 19%) were recorded in England from 2011 to 2016, with no change in the rates of occasional smoking (4% both in 2011 and 2016).[13] Any impact on tobacco use of the recent upsurge in e-cigarette use in children will become more certain as the period of observation is extended. Experimental studies can also provide pertinent evidence.

The limited experimental evidence concerning the impact of e-cigarette exposure on children has focused on exposure to e-cigarette advertising. In one study, children exposed to televised e-cigarette adverts expressed more positive attitudes towards and greater intentions to use e-cigarettes.[14] In another study, children seeing candy-flavoured e-cigarette adverts found these adverts more appealing and were more interested in buying and trying the products when compared with those children exposed to non-flavoured e-cigarette adverts.[15] But in neither study did exposure to e-cigarette advertisements significantly increase the appeal of smoking tobacco cigarettes. Only one study to date has found a cross-product influence of e-cigarette adverts on perceptions of the harms of occasional tobacco smoking.[16] In this study, exposing children to e-cigarette adverts characterised as depicting glamour or health had no significant impact on the appeal of smoking tobacco cigarettes, or the perceived harm of smoking more than 10 cigarettes per day. However, those exposed to either set of adverts perceived the harms of smoking one or two tobacco cigarettes occasionally to be lower than did those not exposed to any adverts.

Even though the size of the effect of perceived risk on routine or habitual behaviours is small to moderate,[17 18] it is nonetheless important in this context given the harms of tobacco smoking. Perceived harm (risk) of occasional smoking predicts tobacco smoking.[19 20] Furthermore, although the health consequences of occasional smoking can be as severe as regular smoking,[21] young smokers who smoke occasionally do not consider themselves smokers, believing they are immune to the risks associated with smoking, and have low intentions to quit.[22 23] In a similar vein, perceived risk significantly predicts intentions and behaviours generally,[17 18] as well as more specifically in relation to smoking, with perceived harm associated with greater likelihood of staying abstinent or quitting if smoker.[24–26]

The aim of the present study is to replicate and extend recent findings showing that children perceive the harms of occasional tobacco smoking to be lower after exposure to e-cigarette adverts. By using a larger sample of children aged 11–16 and a control condition with equivalent task demands in which children were exposed to adverts for objects unrelated to tobacco smoking or vaping (pens), we sought to provide a more robust estimate of the effect found by Petrescu and colleagues.[16] In addition to assessing children's perceptions of the harms of occasional tobacco smoking, the present research also aimed to extend prior literature by examining children's perceptions of the harms of regular tobacco smoking, the perceived normativeness of tobacco smoking, and children's susceptibility to future tobacco smoking. In order to provide a more complete understanding of children's perceptions towards different nicotine products, we adapted all the measures assessing perceptions of tobacco smoking to also assess children's perceptions pertaining to e-cigarette use (including perceived harm, normativeness and potential susceptibility for future use).

## METHODS

### Design

A between-subjects experiment with one independent factor of two levels corresponding to the advertisements to which participants were exposed:

A. Adverts depicting e-cigarette use as glamorous.

B. Adverts for objects (pens) unrelated to tobacco smoking or vaping (control condition).

### Participants

Data were collected from 1449 English school children aged between 11 and 16 years (sampled from three schools, two based in Cambridgeshire and one based in Hampshire). Data were collected and analysed between January and September 2016. Randomisation was successful: there were no significant differences between the two experimental groups on any of the demographic, smoking or e-cigarette use characteristics measured. Ever-smokers and ever-users of e-cigarettes were excluded from the analyses leaving a final sample of 1057 participants. Characteristics of the full and final samples are shown in table 1A and B, respectively. This sample size provided more than 90% power at $\alpha=0.05$ to detect a small-sized effect ($d=0.27$) of glamorous e-cigarette adverts on the perceived harm of occasional tobacco smoking (based on a recent study by Petrescu et al),[16] allowing for a reduction in sample size caused by excluding children with prior tobacco smoking or e-cigarette use.[27]

### Intervention

Each experimental condition displayed 10 adverts, with the control condition showing adverts of pens, and the e-cigarette condition showing adverts associating e-cigarette use with glamour. The e-cigarette adverts were taken from Petrescu et al.[16] The e-cigarette adverts for that study were sampled from the Stanford Adverts Repository (http://tobacco.stanford.edu/tobacco_main/index.php). A subset of 40 possible e-cigarette adverts were pilot tested with 16 year olds. Ten adverts were selected based

**Table 1** (A) Demographic and smoking-related characteristics of all randomised participants (n = 1449). (B) Demographic characteristics of final sample of non-smokers and non-users of e-cigarettes (n = 1057)

**A**

| | E-cigarette adverts (n=714) | Control adverts (n=735) | Test statistic | P values | Total (n=1449) |
|---|---|---|---|---|---|
| Age - M (SD) | 13.71 (1.40) | 13.73 (1.33) | 0.235 | 0.815 | 13.72 (1.37) |
| Gender—male % (n) | 48.5 (346) | 50.1 (368) | 0.933 | 0.334 | 49.3 (714) |
| Ethnicity—White % (n) | 74.6 (533) | 72.9 (536) | 0.557 | 0.456 | 73.8 (1069) |
| Regular cigarette use—Yes % (n) | 12.3 (88) | 12.1 (89) | 0.032 | 0.858 | 12.2 (177) |
| Cigarette experimentation—Yes % (n) | 16.1 (115) | 15.1 (111) | 0.348 | 0.555 | 15.6 (226) |
| E-cigarette awareness—Yes % (n) | 92.9 (663) | 93.9 (690) | 0.157 | 0.692 | 93.4 (1353) |
| E-cigarette use—Yes % (n) | 19.9 (142) | 21.1 (155) | 0.230 | 0.631 | 20.5 (297) |
| Cigarette use first in dual use—% (n) | 8.7 (62) | 7.9 (58) | 0.003 | 0.956 | 8.3 (120) |
| E-cigarette use first in dual use—% (n) | 8.3 (59) | 7.6 (56) | 0.003 | 0.956 | 7.9 (115) |

**B**

| | E-cigarette adverts (n=521) | Control adverts (n=536) | Test statistic | P values | Total (n=1057) |
|---|---|---|---|---|---|
| Age—M (SD) | 13.46 (1.40) | 13.50 (1.34) | 0.472 | 0.637 | 13.48 (1.37) |
| Gender—male % (n) | 45.1 (235) | 48.7 (261) | 3.147 | 0.076 | 46.9 (496) |
| Ethnicity—White % (n) | 74.9 (390) | 73.1 (392) | 0.407 | 0.524 | 74.0 (782) |

For all variables reported above, differences between the experimental groups were examined using $\chi^2$ tests, apart from the age variable which was examined using an independent samples $t$-test.

on ratings for their depiction of glamour (for more details see Petrescu *et al*).[16] The adverts for the control condition were selected from a larger sample of pen adverts. The pen adverts were sourced online. Pen adverts were chosen as the control stimuli due to their similar shape and look to tobacco and e-cigarettes. Three authors (MV, ASJW, SC) selected pen adverts to match the content of the e-cigarette adverts, including the presence of a person (with four adverts showing a woman using a pen, four adverts showing a man using a pen and two adverts with no person in the advert).

### Measures
#### Primary outcome
*Perceived harm of occasional tobacco smoking* was assessed by an item adapted from Wakefield *et al*.[28] 'How dangerous do you think it is to smoke one or two cigarettes occasionally?' rated on a five-point scale, 1=not very dangerous to 5=very dangerous.

#### Secondary outcomes
*Perceived harm of tobacco smoking regularly and in general* was measured using two items.[28] 'Smoking can harm your health' rated from 1=strongly disagree to 5=strongly agree, and 'How dangerous do you think it is to smoke more than 10 cigarettes a day?' rated from 1=not very dangerous to 5=very dangerous. These were analysed separately as in previous studies.[28 29]

*Perceived risk of developing tobacco-related diseases* was measured by items adapted from Pepper *et al*.[30] 'How

likely do you think it is that smoking tobacco cigarettes more than 10 times a day regularly (smoking tobacco cigarettes once or twice occasionally) would cause you to develop each of the following in the next 10 years? (If you're not sure, please give us your best guess) (a) lung cancer, (b) heart disease and (c) mouth or throat cancer'. Ratings were provided on scales from 1=not at all likely to 5=extremely likely. Two separate composite indices were made for perceived risk from regular ($\alpha$=0.76) and occasional ($\alpha$=0.90) tobacco smoking, respectively.

*Prevalence estimates of tobacco smoking* were given on an open-ended question: 'How many young people your age out of 100 do you think smoke tobacco cigarettes?'.[31]

*Susceptibility to tobacco smoking* was measured using three items: 'Do you think you will be smoking tobacco cigarettes when you are 18 years old?'; 'Do you think you will smoke a tobacco cigarette at any time during the next year?' and 'If one of your friends offered you a tobacco cigarette, would you smoke it?'.[32] Participants were categorised as susceptible if they did not respond 'definitely not' to all three items.

*Appeal of adverts* was assessed by asking: 'How much do you like this advert (not the product)?'.[33] Responses ranged from 1=not at all to 4=a lot. Responses to the 10 adverts had high internal consistency ($\alpha$=0.80) and were averaged into a single index.

*Interest in buying and trying products displayed in the adverts* was assessed with the item: 'Does this advert make you want to buy and try this product?' with scores ranging

 

from 1=not at all to 4=yes, a lot.[33] Responses had high internal consistency across the 10 adverts and were averaged into a single index ($\alpha$=0.85).

*Perceptions of e-cigarette use:* All of the outcomes described above, gauging perceptions of tobacco smoking, were adapted to also assess perceptions of using e-cigarettes (including: perceived harm of occasional and regular/general use of e-cigarettes; perceived risk of developing tobacco related diseases by using e-cigarettes regularly/occasionally; prevalence estimates of e-cigarette use; and susceptibility to use e-cigarettes). The composite indices for perceived risk from regular ($\alpha$=0.93) and occasional ($\alpha$=0.95) e-cigarette use had good interitem reliabilities.

### Other measures

Tobacco smoking was measured with two items: 'Have you ever smoked a tobacco cigarette?' and 'Have you ever tried tobacco cigarette smoking, even one or two puffs?'.[32] Items assessing tobacco cigarette smoking were adapted to assess use of e-cigarettes: 'Have you ever used an e-cigarette?' and 'In the past 30 days, on how many days did you use an e-cigarette?' For dual users, we also asked: 'If you are both smoking tobacco cigarettes and using e-cigarettes, which product did you start using first?'. Gender, age and ethnicity were also recorded.

### Procedure

Prior passive parental consent was obtained, and the head teachers of the schools acted *in loco parentis* during data collection. The schools sent parents of eligible children letters to their home addresses and e-mail accounts with the Information Sheet and Opt-out Consent Forms for the present study. Children who were opted out from participating in the study took part in alternative lesson arrangements organised by the schools. Before commencing the study, children also verbally assented to participation. Participating children were then reminded that they could withdraw from the study at any point.

The study materials were presented in paper–pencil format, with each participant receiving a booklet corresponding to one of the two experimental conditions depending on randomisation. Participants in the e-cigarette and control advert conditions were each exposed to a series of 10 print adverts in their booklets. To ensure that participants engaged with the adverts, after each advert, they were asked to rate the appeal of the advert, and their interest in buying and trying the product (see Measures section). Children in both experimental conditions were told that the study was about their views on e-cigarettes and tobacco cigarettes. Children completed the experimental booklets at their own pace, and exposure to the adverts was not timed. The order in which the adverts appeared was fixed across participants. Potential confusion between e-cigarettes and tobacco cigarettes was managed by: (1) presenting all items pertaining to tobacco cigarettes and e-cigarettes in two separate sections; (2) adding a heading at the beginning of each section informing participants that the next section will deal with either tobacco or e-cigarettes; (3) including a picture of a tobacco cigarette and a picture of an e-cigarette at the beginning of each section; and (4) including a definition of e-cigarettes before the presentation of adverts and before assessing e-cigarette-related items.

Participants were randomly assigned to one of the two groups, using a pre-established random sequence generated by the statistical package R. Prior to the testing session, the different versions of the booklets were arranged in the prerandomised order and these booklets were then distributed during testing. Both experimenters and participating children were blinded to allocated randomisation (even though children were exposed to adverts, they only saw one type of advert and were not aware of what kind of adverts the other children were shown). Experimenters made sure that participants finishing earlier than others remained seated until everyone had finished. Once participants had completed their questionnaires, they were provided with a verbal and written debrief about the nature of the study.

### Patient and public involvement

Four children who were the same age as eligible participants were asked to comment on the questionnaire materials prior to testing. The children gave suggestions on how the materials could be edited to make them easier to understand for participating children. The children who piloted the materials were not involved in study recruitment and conduct. Participating children will be informed of the study results with a short summary message distributed via their schools.

### Statistical analysis

All analyses were conducted using SPSS (V.23), R (V.3.3.1) and Review Manager (V.5.3). Responses on the primary and secondary outcomes were not normally distributed. Subsequent analyses were therefore conducted using non-parametric statistical tests (Mann-Whitney U, $\chi^2$ and ordinal regression) to test equality of the location parameter between treatment groups. To provide a summary of the effects of e-cigarette advertising on perceived harm of occasional tobacco smoking, we meta-analysed the present data and the results of two published studies that also examined the impact of different types of e-cigarette adverts on perceptions of tobacco harm.[34] We searched published records for studies that could be synthesised, so the meta-analysis provides an accurate representation of all evidence currently available to us (for more details on the search strategy used and the included/excluded studies for the meta-analysis, please see online supplementary materials). All measures, experimental conditions and sample size calculations are reported in the manuscript. Exploratory analyses were also conducted on the subsample made up of ever-smokers and ever-users of e-cigarettes in order to explore whether e-cigarette adverts will have similar effects in that subsample (please see online supplementary materials). Additional exploratory analyses examined whether age, gender or ethnicity

moderated the effects of experimental condition on the primary outcome of interest (these analyses can be seen in the online supplementary materials).

## RESULTS
### Primary outcome
*Perceived harm of occasional tobacco smoking*: Children exposed to glamorous e-cigarette adverts (mean rank=508.69) perceived the danger as lower than did the control group (mean rank=546.84, Mann-Whitney U=129045.500, $Z$=−2.129, p=0.033). Using ordinal regression (controlling for clustering at the level of school) replicated these results ($t$=−2.131, p=0.033).

### Secondary outcomes
There were no statistically significant differences between the experimental groups in the perceived harm of regular smoking and smoking in general; perceived risk of developing tobacco-related diseases due to regular and occasional smoking; perceived susceptibility to smoking tobacco cigarettes; or the prevalence estimates for tobacco smoking. Similarly, there were no statistically significant differences between the experimental groups in: perceived harm of using e-cigarettes occasionally, regularly or in general; perceived risk of developing tobacco-related diseases due to regular and occasional use of e-cigarettes; perceived susceptibility to using e-cigarettes; or prevalence estimates for using e-cigarettes. Please see table 2 for more details on these analyses.

Children exposed to glamorous e-cigarette adverts (mean rank=426.32) liked the adverts less than did those in the control group (mean rank=628.80, Mann-Whitney U=86133.500, $Z$=−10.797, p<0.001). Furthermore, children exposed to glamorous e-cigarette adverts (mean rank=393.83) were less interested in buying and trying the products shown in the adverts than were those in the control group (mean rank=660.39, Mann-Whitney U=69202.500, $Z$=−14.298, p<0.001).

### Meta-analysis
The same measure of perceived harm of occasional tobacco smoking was used in two other similar studies (see online supplementary materials for more details on the search strategy used to identify eligible studies for synthesis). These assessed the impact of exposure to candy-like flavoured and non-flavoured e-cigarette adverts,[15] and the impact of glamorous and healthful e-cigarette adverts.[16] Using results from these two studies and the current study, we conducted a meta-analysis (using Review Manager V.5.3) of the continuous outcome, comparing those exposed to any type of advert for e-cigarettes with those in the control groups.

Exposing children to adverts for e-cigarettes decreased their perceived harm of occasional tobacco smoking: SMD=−0.15, 95% CI −0.24 to −0.06, $I^2$=48%, $Z$=3.21, p=0.001 (see figure 1). Similar results were obtained when dichotomising responses to this outcome (as in Petrescu *et al*).[16]

| Outcome variable | E-cigarette adverts (n=521) | Control adverts (n=536) | Test statistic | P values |
|---|---|---|---|---|
| Perceived harm of occasional tobacco smoking | 508.69 | 546.84 | −2.129 | 0.033 |
| Perceived harm of tobacco smoking in general | 525.10 | 529.84 | −0.435 | 0.664 |
| Perceived harm of regular tobacco smoking | 531.91 | 524.18 | −0.512 | 0.609 |
| Perceived disease risk (regular smoking) | 529.87 | 522.22 | −0.415 | 0.678 |
| Perceived disease risk (occasional smoking) | 540.36 | 512.05 | −1.524 | 0.127 |
| Tobacco smoking prevalence estimates | 521.96 | 513.12 | −0.477 | 0.634 |
| Susceptibility to tobacco smoking | 42.4 | 37.9 | 2.515 | 0.113 |
| Perceived harm of occasional e-cigarette use | 527.49 | 530.47 | −0.167 | 0.867 |
| Perceived harm of e-cigarette use in general | 516.81 | 539.84 | −1.282 | 0.200 |
| Perceived harm of regular e-cigarette use | 530.06 | 527.97 | −0.116 | 0.908 |
| Perceived disease risk (regular e-cigarette use) | 520.34 | 527.56 | −0.389 | 0.697 |
| Perceived disease risk (occasional e-cigarette use) | 523.22 | 526.74 | −0.193 | 0.847 |
| E-cigarette use prevalence estimates | 523.19 | 513.90 | −0.501 | 0.616 |
| Susceptibility to e-cigarette use | 50.1 | 49.8 | 0.015 | 0.902 |
| Appeal of adverts | 426.32 | 628.80 | −10.797 | <0.001 |
| Interest in buying and trying advertised product | 393.83 | 660.39 | −14.298 | <0.001 |

For all outcome variables, the test statistic corresponds to the $Z$ value from the Mann-Whitney U analyses (with corresponding mean ranks shown for each experimental group), except for the variables susceptibility to smoking and e-cigarettes use which are binary variables and are denoted by percentages summarised using the $X^2$ test statistic.

| Study or Subgroup | Experimental Mean | SD | Total | Control Mean | SD | Total | Weight | Std. Mean Difference IV, Fixed, 95% CI | Year | Std. Mean Difference IV, Fixed, 95% CI |
|---|---|---|---|---|---|---|---|---|---|---|
| Vasiljevic et al., 2016 | 2.99 | 1.12 | 313 | 3.07 | 1.05 | 156 | 22.8% | -0.07 [-0.26, 0.12] | 2016 | |
| Petrescu et al., 2017 | 3.18 | 1.23 | 278 | 3.57 | 1.03 | 133 | 19.5% | -0.33 [-0.54, -0.13] | 2017 | |
| Vasiljevic et al., 2018 | 2.85 | 0.97 | 521 | 2.97 | 1.04 | 534 | 57.7% | -0.12 [-0.24, 0.00] | 2018 | |
| **Total (95% CI)** | | | **1112** | | | **823** | **100.0%** | **-0.15 [-0.24, -0.06]** | | |

Heterogeneity: Chi² = 3.84, df = 2 (P = 0.15); I² = 48%
Test for overall effect: Z = 3.21 (P = 0.001)

**Figure 1** Forest plot of meta-analysis of impact of exposure to e-cigarette adverts on the perception that occasional smoking of one or two cigarettes is not very dangerous (continuous outcome).

## Discussion

Children exposed to e-cigarette adverts depicting glamour perceived the harms of smoking one or two tobacco cigarettes occasionally to be lower than did those exposed to unrelated adverts. These results corroborate previous findings.[16] An updated meta-analysis comprising three studies (including the present study) with 1935 children confirmed that exposure to different types of e-cigarette adverts (glamorous, healthful, flavoured, or non-flavoured) lowers the perceived harm of occasional smoking of one or two tobacco cigarettes. The current study also replicates previous findings that exposure to glamorous and other types of adverts does not affect children's perceptions of the (high) harm of regularly smoking more than 10 tobacco cigarettes per day.[15 16] Our findings suggest that exposure to adverts for e-cigarettes may lead to differences in how children perceive the harms of tobacco smoking.

The absence of a significant impact of viewing e-cigarette adverts on perceptions of the harms associated with regularly smoking more than 10 tobacco cigarettes a day is encouraging (see also[15 16]). However, the impact on perceived harms of occasional smoking is concerning given that such perceptions can predict subsequent smoking.[19 20] Young occasional smokers in particular do not consider themselves smokers, believing they are immune to the risks associated with smoking, with low intentions to quit.[22 23] The effect of e-cigarette adverts on perceived harms of occasional tobacco smoking is therefore both theoretically and empirically important, given that perceived harm (risk) is a key construct affecting health behaviour change in multiple theories of behaviour change (see[35]). Furthermore, the observed differential effects on the perceived harms of occasional versus regular smoking may provide an indication that the former behaviour may be easier to mentally 'justify', thereby providing another potential route to self-regulation failure.[36]

Interestingly, children perceived that the harm of occasional tobacco smoking was lower when they were exposed to e-cigarette adverts, even though they rated the e-cigarette adverts as significantly less appealing and professed a lower interest in buying and trying the e-cigarettes when compared with the pens shown in the control condition. These findings may have important ramifications for future research and policy, since they suggest that the cross-product impact of e-cigarette adverts may largely work via an unconscious, implicit route that may not necessarily affect self-reported explicit appeal, but may change perceptions of harm (risk) which feed into children's behavioural decisions. These hypotheses merit further testing.

In more general terms, the population consequences of our findings are currently unknown. Two sets of outcomes need to be considered. First, the possible impact on tobacco smoking and second the possible impact on attitudes towards tobacco smoking. First, a small change in perceived harm of occasional smoking and no change in the already high perceived harm of smoking 10 or more cigarettes on a regular basis may have no impact on the likelihood that children smoke tobacco cigarettes. This is supported by the evidence that perceived harms of occasional tobacco smoking have a small to moderate effect on actual smoking.[19 20] It is also consistent with the evidence that despite exposure to adverts and vaping, there is no corresponding increase in the overall rates of children smoking tobacco. Indeed, the decline in rates observed over the last two decades has continued.[13 27 37] Nonetheless, any impact of e-cigarette adverts on tobacco smoking in children demands attention from policy makers.

Second, a lower perceived harm of occasional smoking may lead to more positive attitudes towards tobacco smoking and the tobacco industry, which in turn may result in more negative attitudes towards tobacco control policies. In high-income countries, public attitudes towards tobacco control policies, particularly those targeting children, are currently very positive.[38 39] Such attitudes are important in supporting policy makers in implementing effective tobacco control policies. Any lessening of these positive attitudes towards tobacco control would be a concern.

### Strengths and limitations with future directions

The large sample of children and the use of a control condition in which children were exposed to a battery of adverts of objects unrelated to tobacco cigarettes or e-cigarettes strengthen the conclusions that can be drawn from the present study. By using a control condition in which children were exposed to pen adverts, we were able to isolate the effects of e-cigarette adverts, and conclude that findings of lowered harm of occasional tobacco smoking can be attributed to e-cigarette adverts and not to viewing adverts more generally. Another strength of the current study is its contribution to an updated meta-analysis providing the most robust evidence to date that e-cigarette adverts of different kinds (glamorous, healthful,

flavoured, or non-flavoured) may have a cross-product influence in lowering children's perceptions of the harms of occasional tobacco smoking.

The study was limited in several respects. The primary outcome was a belief and not a behaviour. Future studies should examine whether perceptions of harm following exposure to e-cigarettes translate into actual smoking behaviour.

The between-subjects design allowed us to control for any possible carry-over effects of the different types of adverts. But this design also limits our ability to account for baseline differences in susceptibility to future tobacco smoking. Future research might usefully incorporate within-subjects designs or assess baseline levels of susceptibility to tobacco smoking which could be controlled for in subsequent analyses.

The study was further limited in assessing the impact of momentary exposure to e-cigarette adverts. The results may therefore provide an underestimation of the true effects of e-cigarette advertising which is more dynamic and pervasive in everyday settings (eg, billboards, posters, internet). Future research should examine other forms of e-cigarette advertising, and use a longitudinal design to corroborate the present findings. Further research is also warranted on the link between exposure to e-cigarette adverts, attitudes towards the tobacco industry and support for tobacco control policies.

Field experiments would provide a useful complement to the present study, since it is unclear whether the present findings obtained via a survey administered in schools are generalisable to the real world. Furthermore, it is possible that the adverse effects of e-cigarette advertising found in this study may be short-lived. Whether short exposure to e-cigarette adverts has long-term effects on perceived harms of occasional tobacco smoking can only be ascertained by assessing outcomes in the longer as well as shorter term.

### Policy implications

Our findings suggest that policies regarding e-cigarette advertising need to take into account the potential adverse cross-cueing effects on tobacco smoking among children. The present study coupled with two previous studies that have examined perceptions of the harms of tobacco smoking following exposure to e-cigarette adverts among children suggests the need to re-examine current regulations on advertising. E-cigarette advertising in the European Union (EU) is currently subsumed under the new Tobacco Products Directive (TPD).[40] These recent regulations limit the exposure of children to TV and newspaper e-cigarette advertising. However, the implementation of these regulations across EU member states still allows some form of e-cigarette advertising (posters, leaflets, billboards in shops), so children are still exposed to e-cigarette adverts. The TPD also does not explicitly prohibit the use of advertising themes/content that may be particularly appealing to children (such as flavoured, or glamorous e-cigarette adverts). Likewise, in the USA,

the Food and Drug Administration recently began regulating e-cigarettes, but these regulations do not include provisions to curb children's exposure to e-cigarette advertising or to restrict e-cigarette adverts with potentially youth-appealing themes/content.[41]

## CONCLUSIONS

This study adds to existing evidence that exposure to e-cigarette adverts reduces children's perceptions of the harm of occasional tobacco smoking. Further studies are warranted, using longitudinal and experimental designs, to assess a wider range of possible impacts of the marketing of e-cigarettes including attitudes towards the tobacco industry and tobacco control policies.

**Acknowledgements** We would like to thank Emma Cartwright, Catherine Galloway and Zorana Zupan for their assistance with data collection. We would also like to thank all participating children and the children who piloted the materials, as well as the teachers who assisted with the project.

**Contributors** MV supervised the study and oversaw the acquisition of data. MV and D-LC were responsible for the data analysis. MV drafted the manuscript. ASW, SC, D-LC, SS and TMM provided critical revisions to the manuscript. All authors collaborated in designing the study, contributed to the interpretation of results and read and approved the final version of the manuscript.

**Funding** This report is an independent research commissioned and funded by the National Institute for Health Research Policy Research Programme (Policy Research Unit in Behaviour and Health (PR-UN-0409-10109)).

**Disclaimer** The views expressed in this publication are those of the authors and not necessarily those of the NHS, the National Institute for Health Research, the Department of Health and Social Care or its arm's-length bodies and other government departments. The final version of the report and ultimate decision to submit for publication was determined by the authors.

**Competing interests** None declared.

**Patient consent** Not required.

**Ethics approval** University of Cambridge's Psychology Research Ethics Committee [PRE.2015.106].

**Provenance and peer review** Not commissioned; externally peer reviewed.

**Data sharing statement** We are willing to make all data available to any interested parties. Please contact the corresponding author for more information.

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
