## [Reviewer comments · BMJ Open]

ARTICLE DETAILS

TITLE (PROVISIONAL)	E-cigarette adverts and children's perceptions of tobacco smoking harms: An experimental study and meta-analysis
AUTHORS	Vasiljevic, Milica; St John Wallis, Amelia; Codling, Saphsa; Couturier, Dominique-Laurent; Sutton, Stephen; Marteau, Theresa

VERSION 1 – REVIEW

REVIEWER	Mark Conner University of Leeds, UK
REVIEW RETURNED	08-Nov-2017

GENERAL COMMENTS	This is an interesting study adding to our understanding of the impact of e-cigarette advertising and views of the risk of smoking and particularly occasional smoking using an experimental design. A number of issues might usefully be addressed in a revision. 1. A key issue is the decision to drop any adolescents who currently use e-cigarettes or who currently smoke. It would be interesting to report any effects of advertising in this particular group although I agree with the authors that a focus on never users of e-cigarettes or cigarettes is useful. The growing focus of research on the link between e-cigarette use and escalation of smoking means the impacts of adverts in those who have tried e-cigarettes or smoking may be of interest.2. Abstract. I found the description of the outcomes unclear. The use of the term endpoints is confusing. It needs to be made clearer that the same set of questions were used in relation to cigarettes and e-cigarettes.3. Strengths. Here and in the discussion the reference to tobacco control policies needs to be revised (presumably it could lead to more negative attitudes to tobacco control policies?).4. It would be useful if the introduction distinguished between rates of regular and occasional (or even ever) use of cigarettes given the current focus is on lower levels of smoking.5. Method. In relation to the sample size calculation it would be useful to specify the small effect size used. I was also unsure about how to interpret the alpha for ratings of adverts? It would be useful to specify the items used in relation to e-cigarettes. The justification of what were primary and secondary outcomes was unclear.6. Results. These do not seem to map closely on to Tables 2 and 3. I was not clear why the table of means were reported if the data was skewed? The text needs to refer to the lack of effects and reference the tables for the exact findings.7. Discussion. A further weakness worth acknowledging was the fact that these effects may be short-lived and disentangling effects of adverts seen in the lab versus seen in the real world may be difficult.
---

REVIEWER	Adam Cole School of Public Health and Health Systems, University of Waterloo, Waterloo, Ontario, Canada
REVIEW RETURNED	19-Jan-2018

GENERAL COMMENTS	This study replicates existing research for the association between exposure to e-cigarette advertising and perceptions of harm of tobacco smoking and e-cigarette use among a cross-sectional sample of students in the UK. The current study employs a stronger design than previous works with the use of a strong control condition that mimics e-cigarette advertising and a larger sample size. The manuscript is very well-written and organized. The authors provide a clear and concise introduction that illustrates the hypothesized link between exposure to e-cigarette advertising and subsequent intentions to smoke cigarettes. The design and methods are sound and clearly explained and follow a similar approach to previous works, allowing for comparisons to be made to previous studies. Some revisions are suggested to clarify the results and the inclusion of a meta-analysis. Results: It would be helpful to include the results of statistical tests (e.g., chi-square and p-values) comparing demographic and smoking-related characteristics of all randomized participants and the final sample in Tables 1(a) and 1(b) to show that characteristics of participants in the experimental groups were the same. Results: It would be helpful to include the results of statistical tests (e.g., z scores, t scores, p-values) comparing outcome measures across experimental groups in Tables 2 and 3. Results: Consider testing for differences between experimental groups in susceptibility to tobacco smoking and susceptibility to e-cigarette use using a chi-square test or test of proportion. It appears as though there could be a significant difference in the percentage of children susceptible to tobacco smoking between experimental groups. Results: It's helpful that the authors included a meta-analysis of the impact of exposure to e-cigarette advertising on the perception of harms from occasional smoking. However, additional information should be included about the search strategy, study selection, data collection process, etc. to strengthen the results and ensure that all relevant studies and data have been included. Consider using a PRISMA Checklist to ensure all the necessary components have been included and sufficient detail has been provided. Results: One benefit of the study sample is that there are children that report ever smoking tobacco cigarettes and e-cigarettes (although they were excluded from the current analyses). Given that exposure to e-cigarette advertisements could have a different impact on ever smokers (relative to non-smokers) it may be worth exploring whether similar results are identified among the (small) sample of ever smokers and e-cigarette users (e.g., lower perceived harm of occasional or regular tobacco smoking following exposure to e-cigarette advertisements). Such results could provide additional justification for limiting e-cigarette advertising. Discussion: it's interesting and surprising that the appeal and interest in buying and trying the advertised products was actually significantly lower in the e-cigarette advertisement group relative to
--

	the control condition. It would be worth commenting in the discussion why this might be and in what way it might have impacted the results of the study. Discussion: The between-subjects design of the study is an additional limitation. A within-subjects design would have been stronger and could have accounted for baseline differences in susceptibility to future smoking.
--	---

VERSION 1 – AUTHOR RESPONSE

Reviewers' Comments to Author:

Reviewer: 1

Reviewer Name: Mark Conner

Institution and Country: University of Leeds, UK

Competing Interests: None

This is an interesting study adding to our understanding of the impact of e-cigarette advertising and views of the risk of smoking and particularly occasional smoking using an experimental design. A number of issues might usefully be addressed in a revision.

- We are very pleased with your positive assessment of our work, and hope that we can address all points for improvement that you noted in your review.

1. A key issue is the decision to drop any adolescents who currently use e-cigarettes or who currently smoke. It would be interesting to report any effects of advertising in this particular group although I agree with the authors that a focus on never users of e-cigarettes or cigarettes is useful. The growing focus of research on the link between e-cigarette use and escalation of smoking means the impacts of adverts in those who have tried e-cigarettes or smoking may be of interest.

- We have followed your suggestion (and that of Reviewer 2) and conducted these additional Exploratory Analyses on the subsample of ever-smokers and ever-users of e-cigarettes. These analyses together with a comprehensive table of results (Table S1) can be found in Online Supplementary Materials. We are very cautious in presenting these analyses since they were not pre-planned and likely statistically underpowered given the sample size of ever-smokers and ever-users of e-cigarettes is likely to be too small and inadequate to provide correct estimates of the effect sizes of the different analyses in this subgroup. Nevertheless, we agree that by providing these exploratory analyses in the Online Supplementary Materials, they may provide useful information for future research in this area.

2. Abstract. I found the description of the outcomes unclear. The use of the term endpoints is confusing. It needs to be made clearer that the same set of questions were used in relation to cigarettes and e-cigarettes.

- Many thanks for encouraging us to clarify our outcome measures in the Abstract section. Revisions can be seen below and on page 2 of the revised manuscript:

<< Outcomes: Perceived harm of occasional smoking of one or two tobacco cigarettes was the primary outcome. Secondary outcomes included: perceived harm of regular tobacco smoking, susceptibility to tobacco smoking and perceived prevalence of tobacco smoking in

young people. Perceptions of using e-cigarettes were gauged by adapting all the outcome measures used to assess perceptions of tobacco smoking. >>

3. Strengths. Here and in the discussion the reference to tobacco control policies needs to be revised (presumably it could lead to more negative attitudes to tobacco control policies?).

- Following recommendation by the Editor that the 'Strengths and Limitations' section that appears just after the Abstract should only contain bullet points that relate to the methods or design of the study we have now deleted the reference to implications of our findings.
- We thank you for spotting the discrepancy in how we described the potential impact of our results on tobacco control policies in the Discussion section. We have tried to clarify this on page 17 of the revised Discussion:

<< Second, a lower perceived harm of occasional smoking may lead to more positive attitudes towards tobacco smoking and the tobacco industry, which in turn may result in more negative attitudes towards tobacco control policies. In high income countries, public attitudes towards tobacco control policies, particularly those targeting children, are currently very positive.^{38,39} Such attitudes are important in supporting policy makers in implementing effective tobacco control policies. Any lessening of these positive attitudes towards tobacco control would be a concern. >>

4. It would be useful if the introduction distinguished between rates of regular and occasional (or even ever) use of cigarettes given the current focus is on lower levels of smoking.

- We have now expanded the Introduction section to include figures on ever and occasional tobacco smoking. Please note that these figures pertain only to England, since comparative data do not exist from the US where only regular tobacco smoking prevalence is commonly reported. Please see below and pages 4-5 of the revised manuscript for our edits:

<< Several prospective studies in the USA and UK have found that among children e-cigarette use predicts tobacco smoking one year later.⁹⁻¹³ By contrast, population level data show that the rising use and experimentation of e-cigarettes among children is accompanied by a continued decline in regular tobacco smoking in that group, from 15.8% to 9.2% amongst US high-schoolers in the period from 2011 to 2014,³ and from 5% in 2010 to 3% in 2014 amongst 11-15 year olds in England.¹⁴ Similar declines in rates of occasional (4% to 2%) and ever smoking tobacco (25% to 18%) were recorded in England from 2010 to 2014.¹⁴ Any impact on tobacco use of the recent upsurge in e-cigarette use in children will become more certain as the period of observation is extended. Experimental studies can also provide pertinent evidence. >>

5. Method. In relation to the sample size calculation it would be useful to specify the small effect size used. I was also unsure about how to interpret the alpha for ratings of adverts? It would be useful to specify the items used in relation to e-cigarettes. The justification of what were primary and secondary outcomes was unclear.

- We have now specified the small effect size used for the sample size calculations. Please see page 7 of the revised manuscript:

<< This sample size provided more than 90% power at $\alpha = .05$ to detect a small-sized effect ($d = 0.27$) of glamorous e-cigarette adverts upon the perceived harm of occasional tobacco smoking (based on a recent study by Petrescu et al),¹ allowing for a reduction in sample size caused by excluding children with prior tobacco smoking or e-cigarette use. >>

- We have reworded our description regarding Cronbach's alphas for the ratings of the 10 adverts. Please see page 10 and below:

<< Appeal of adverts was assessed by asking: "How much do you like this advert (not the product)?"³³ Responses ranged from 1 = Not at all, to 4 = A lot. Responses to the 10 adverts had high internal consistency ($\alpha = .80$) and were averaged into a single index.

Interest in buying and trying products displayed in the adverts was assessed with the item: "Does this advert make you want to buy and try this product?" with scores ranging from 1 = Not at all, to 4 = Yes, a lot.³³ Responses had high internal consistency across the 10 adverts and were averaged into a single index ($\alpha = .85$). >>

- We have now delineated in greater detail the items used for assessing perceptions of e-cigarette use. Please see page 10:

<< Perceptions of e-cigarette use: All of the outcomes described above, gauging perceptions of tobacco smoking, were adapted to also assess perceptions of using e-cigarettes (including: perceived harm of occasional and regular/general use of e-cigarettes; perceived risk of developing tobacco related diseases by using e-cigarettes regularly/occasionally; prevalence estimates of e-cigarette use; and susceptibility to use e-cigarettes). The composite indices for perceived risk from regular ($\alpha = .93$) and occasional ($\alpha = .95$) e-cigarette use had good inter-item reliabilities. >>

- In the revised Introduction section we now also delineate in further detail the aims of the study, and the choice of primary and secondary outcomes. See page 6:

<< The aim of the present study is to replicate and extend recent findings showing that children perceive the harms of occasional tobacco smoking to be lower after exposure to e-cigarette adverts. By using a larger sample of children aged 11-16 and a control condition with equivalent task demands in which children were exposed to adverts for objects unrelated to tobacco smoking or vaping (pens), we sought to provide a more robust estimate of the effect found by Petrescu and colleagues.¹ In addition to assessing children's perceptions of the harms of occasional tobacco smoking, the present research also aimed to extend prior literature by examining children's perceptions of the harms of regular tobacco smoking, the perceived normativeness of tobacco smoking, and children's susceptibility to future tobacco smoking. In order to provide a more complete understanding of children's perceptions towards different nicotine products, we adapted all the measures assessing perceptions of tobacco smoking to also assess children's perceptions pertaining to e-cigarette use (including perceived harm, normativeness and potential susceptibility for future use). >>

6. Results. These do not seem to map closely on to Tables 2 and 3. I was not clear why the table of means were reported if the data was skewed? The text needs to refer to the lack of effects and reference the tables for the exact findings.

- We have now revised the text to more closely match the information presented in Table 2 (see pages 13-14 and below). Following recommendations, we have now removed Table 3 showing the Means and SDs.

<< Secondary outcomes

There were no statistically significant differences between the experimental groups in the perceived harm of regular smoking and smoking in general; perceived risk of developing tobacco-related diseases due to regular and occasional smoking; perceived susceptibility to smoking tobacco cigarettes; or the prevalence estimates for tobacco smoking. Similarly, there were no statistically significant differences between the experimental groups in: perceived harm of using e-cigarettes occasionally, regularly, or in general; perceived risk of developing tobacco-related diseases due to regular and occasional use of e-cigarettes; perceived susceptibility to using e-cigarettes; or prevalence estimates for using e-cigarettes. Please see Table 2 for more details on these analyses. >>

7. Discussion. A further weakness worth acknowledging was the fact that these effects may be short-lived and disentangling effects of adverts seen in the lab versus seen in the real world may be difficult.

- We have now discussed these additional limitations on pages 17-19 of our revised manuscript:

<< Field experiments would provide a useful complement to the present study, since it is unclear whether the present findings obtained via a survey administered in school are generalisable to the real world. Furthermore, it is possible that the adverse effects of e-cigarette advertising found in this study may be short-lived. Whether short exposure to e-cigarette adverts has long-term effects on perceived harms of occasional tobacco smoking can only be ascertained by assessing outcomes in the longer as well as shorter-term. >>

Reviewer: 2

Reviewer Name: Adam Cole

Institution and Country: School of Public Health and Health Systems, University of Waterloo, Waterloo, Ontario, Canada

Competing Interests: None declared

This study replicates existing research for the association between exposure to e-cigarette advertising and perceptions of harm of tobacco smoking and e-cigarette use among a cross-sectional sample of students in the UK. The current study employs a stronger design than previous works with the use of a strong control condition that mimics e-cigarette advertising and a larger sample size. The manuscript is very well-written and organized. The authors provide a clear and concise introduction that illustrates the hypothesized link between exposure to e-cigarette advertising and subsequent intentions to smoke cigarettes. The design and methods are sound and clearly explained and follow a similar approach to previous works, allowing for comparisons to be made to previous studies. Some revisions are suggested to clarify the results and the inclusion of a meta-analysis.

- We appreciate your positive assessment of our work, and hope that we can address all points for improvement that you noted in your review.

Results: It would be helpful to include the results of statistical tests (e.g., chi-square and p-values) comparing demographic and smoking-related characteristics of all randomized participants and the final sample in Tables 1(a) and 1(b) to show that characteristics of participants in the experimental groups were the same.

- Many thanks for encouraging us to provide the information regarding the exact significance levels of the statistical modelling comparing demographic and smoking-related characteristics shown in Tables 1(a) and 1(b). We have now added extra columns in both these tables, and provide the exact test statistic and the associated p-values of the statistical tests carried out. Please see revised Tables on pages 25-26 of the revised manuscript.

Results: It would be helpful to include the results of statistical tests (e.g., z scores, t scores, p-values) comparing outcome measures across experimental groups in Tables 2 and 3.

- We have now included the relevant statistical tests and p-values in Table 2 to denote exact significance levels for all statistical tests carried out. Please note that following recommendations by Reviewer 1 we have removed Table 3 from the revised version of the manuscript.

Results: Consider testing for differences between experimental groups in susceptibility to tobacco smoking and susceptibility to e-cigarette use using a chi-square test or test of proportion. It appears as though there could be a significant difference in the percentage of children susceptible to tobacco smoking between experimental groups.

- We have tested the indices of susceptibility to tobacco smoking and susceptibility to using e-cigarette with Chi-Squared tests. Outcomes of the Chi-Squared tests were not-significant (more details are now included in Table 2, incl. test statistic and p-values).

Results: It's helpful that the authors included a meta-analysis of the impact of exposure to e-cigarette advertising on the perception of harms from occasional smoking. However, additional information should be included about the search strategy, study selection, data collection process, etc. to strengthen the results and ensure that all relevant studies and data have been included. Consider using a PRISMA Checklist to ensure all the necessary components have been included and sufficient detail has been provided.

- Many thanks for encouraging us to provide more details on the search strategy used to identify relevant studies to be included in the meta-analysis. We have now provided all the details of the search strategies used in the Online Supplementary Materials. We have also attached a PRISMA checklist.

Results: One benefit of the study sample is that there are children that report ever smoking tobacco cigarettes and e-cigarettes (although they were excluded from the current analyses). Given that exposure to e-cigarette advertisements could have a different impact on ever smokers (relative to non-smokers) it may be worth exploring whether similar results are identified among the (small) sample of ever smokers and e-cigarette users (e.g., lower perceived harm of occasional or regular tobacco smoking following exposure to e-cigarette advertisements). Such results could provide additional justification for limiting e-cigarette advertising.

- Many thanks for encouraging us to conduct these additional exploratory analyses. Reviewer 1 had a similar comment (comment [1]). We have now reported these analyses under the heading Exploratory Analyses in the Online Supplementary Materials, to denote that these analyses were not part of our study protocol. For the results of these analyses please see pages 2-3 of the Online Supplementary Materials.

Discussion: it's interesting and surprising that the appeal and interest in buying and trying the advertised products was actually significantly lower in the e-cigarette advertisement group relative to the control condition. It would be worth commenting in the discussion why this might be and in what way it might have impacted the results of the study.

- We have now discussed this finding in our discussion, and offered potential reasons for this pattern of results. Please see page 16 of the Discussion section:

<< Interestingly, children perceived that the harm of occasional tobacco smoking was lower when they were exposed to e-cigarette adverts, even though they rated the e-cigarette adverts as significantly less appealing and professed a lower interest in buying and trying the e-cigarettes when compared to the pens shown in the control condition. These findings may have important ramifications for future research and policy, since they suggest that the cross-product impact of e-cigarette adverts may largely work via an unconscious, implicit route that may not necessarily affect self-reported explicit appeal, but may change perceptions of harm (risk) which feed into children's behavioural decisions. These hypotheses merit further testing.
>

Discussion: The between-subjects design of the study is an additional limitation. A within-subjects design would have been stronger and could have accounted for baseline differences in susceptibility to future smoking.

- Many thanks for encouraging us to discuss this in the Discussion section. Please see page 18 of the revised manuscript:

<< The between-subjects design allowed us to control for any possible carry over effects of the different types of adverts. But this design also limits our ability to account for baseline

differences in susceptibility to future tobacco smoking. Future research might usefully incorporate within-subjects designs or assess baseline levels of susceptibility to tobacco smoking which could be controlled for in subsequent analyses. >>

VERSION 2 – REVIEW

REVIEWER	Mark Conner University of Leeds, UK
REVIEW RETURNED	08-Mar-2018
GENERAL COMMENTS	The authors have done a great job of revising the manuscript and I am happy to recommend acceptance. I have two remaining comments. 1. Given the issue over this being a fast changing field it is disappointing to not include more recent data on adolescent smoking and e-cigarette use. 2016 data is now available from similar sources to those cited for the UK. 2. It would be interesting to know if the effects observed are moderated by age, gender, or ethnicity. For example, is it the case that the reduced risk effect is limited to older adults. This could be reported in supplementary materials.

VERSION 2 – AUTHOR RESPONSE

Reviewer's Comments to Author:

Reviewer: 1

Reviewer Name: Mark Conner

Institution and Country: University of Leeds, UK

Competing Interests: None declared.

The authors have done a great job of revising the manuscript and I am happy to recommend acceptance. I have two remaining comments.

- We appreciate your positive assessment of our work.

1. Given the issue over this being a fast changing field it is disappointing to not include more recent data on adolescent smoking and e-cigarette use. 2016 data is now available from similar sources to those cited for the UK.

- We have now updated all the figures regarding tobacco smoking and e-cigarette use in the USA and Great Britain (or, in England where rates of occasional and ever-smoking amongst adolescents is available), by using the latest figures available from 2016. Please see below and pages 4-5 of the revised manuscript for our edits:

<< The availability and use of e-cigarettes has risen rapidly in the last six years with an estimated 12%-24% of children aged 11-18 experimenting at least once with e-cigarettes in Great Britain in 2015/16,² and 13.5% of middle schoolers and 37.7% of high schoolers in the USA in 2016.^{3,4} >>

<< By contrast, population level data show that the rising use and experimentation of e-cigarettes among children is accompanied by a continued decline in regular tobacco smoking in that group, from 15.8% to 8% amongst US high-schoolers and from 4.3% to 2.2% amongst US middle schoolers in the period from 2011 to 2016,³ and from 5% in 2011 to 3% in 2016

amongst 11-15 year olds in England.¹⁴ Similar declines in rates of ever smoking tobacco (25% to 19%) were recorded in England from 2011 to 2016, with no change in the rates of occasional smoking (4% both in 2011 and 2016).¹⁴ >>

2. It would be interesting to know if the effects observed are moderated by age, gender, or ethnicity. For example, is it the case that the reduced risk effect is limited to older adults. This could be reported in supplementary materials.

- We have now carried out additional moderation analyses by carrying out multiple ordinal regressions. These are now reported in Online Supplementary Materials under Appendix A: Exploratory Analyses (pages 2-5). An extract of our description of these analyses in the Online Supplementary Materials is provided below:

<< We carried out two sets of exploratory analyses. First, multiple ordinal regressions (non-parametric) were conducted to examine any potential interactions between the experimental groups and age, gender, or ethnicity on the primary outcome (perceived harm of occasional tobacco smoking). The results of these ordinal regressions are reported in Tables S1 to S3 below. None of these three demographic variables moderated the effect of experimental group on perceived harm of occasional tobacco smoking (ps > .05). >>